# CTRP6 in Cancer: Mechanistic Insights and Therapeutic Potential

**DOI:** 10.3390/cancers17213409

**Published:** 2025-10-23

**Authors:** Muhammad Zubair Mehboob, Xia Lei

**Affiliations:** Department of Biochemistry and Molecular Biology, Oklahoma State University, Stillwater, OK 74078, USA; mzubair.mehboob@okstate.edu

**Keywords:** CTRP6, cancer progression, therapeutic target

## Abstract

**Simple Summary:**

C1q/TNF-related protein 6 (CTRP6) is a protein that plays important roles in how cancer develops and progresses. It can influence blood vessel formation, cell growth, cell survival, spread of cancer cells, and resistance to treatment. Recent studies also show that CTRP6 may help cancer cells escape a special type of cell death called ferroptosis, which has become an exciting target for new therapies. Because CTRP6 appears to work differently depending on the cancer type, it may serve as both a useful marker for disease and a potential treatment target. In this review, we bring together current knowledge about how CTRP6 affects cancer, highlight the key signaling pathways it controls, and discuss how targeting this protein could open new directions for therapy. Our goal is to provide a clear overview of CTRP6 in cancer to guide future research and therapeutic strategies.

**Abstract:**

C1q/TNF-related protein 6 (CTRP6) is emerging as a critical regulator of cancer biology with direct implications for clinical outcomes. Across a wide spectrum of malignancies, CTRP6 plays a central role in coordinating key oncogenic processes and linking metabolic, inflammatory, and signaling pathways that drive tumor progression. While CTRP6 generally promotes oncogenic behavior in cancers such as hepatocellular carcinoma, lung cancer, and clear cell renal cell carcinoma, conflicting findings have been reported in gastric cancer and oral or head and neck squamous cell carcinoma, where its tumor-promoting versus tumor-suppressive roles remain unresolved. CTRP6 has been shown to modulate fundamental processes including angiogenesis, ferroptosis, proliferation, apoptosis, migration, invasion, and inflammation. These effects are primarily mediated through activation of the PI3K/AKT and MEK/ERK signaling pathways, which are central to tumor growth, metastasis, and therapeutic resistance. Beyond its mechanistic roles, CTRP6 demonstrates potential as a diagnostic and prognostic biomarker, with altered expression patterns linked to cancer initiation, progression, and patient survival. Inhibition of CTRP6 in preclinical models enhances ferroptotic cell death and suppresses tumor progression, highlighting its promise as a therapeutic target. By consolidating current evidence from multiple cancer models, this review provides a comprehensive overview of CTRP6’s contributions to oncogenesis and underscores its dual potential as both a biomarker and a therapeutic target. Advancing a deeper understanding of CTRP6 in specific tumor contexts will be critical for unlocking its clinical utility and may open new opportunities to improve diagnosis, optimize therapeutic strategies, and ultimately enhance patient outcomes.

## 1. Introduction

Cancer remains one of the leading causes of death worldwide, despite significant advances in diagnosis and therapy. Conventional treatments such as surgery, chemotherapy, and radiotherapy have improved patient survival, yet outcomes remain poor for many malignancies due to late detection, tumor heterogeneity, drug resistance, and disease recurrence [1,2]. Targeted therapies and immunotherapies have offered new hope, but their effectiveness is often curtailed by adaptive resistance mechanisms and the immunosuppressive tumor microenvironment (TME) [3,4]. In recent years, nanomedicine has emerged as a promising complementary approach: nanoparticle—based delivery systems can enhance drug accumulation in tumors, reduce systemic toxicity, and help overcome resistance barriers such as poor drug penetration and off—target effects [5,6]. However, even with advanced delivery systems, the success of nanomedicine ultimately hinges on identifying molecular targets that drive tumorigenesis and resistance. Together, these challenges underscore the urgent need to discover and validate novel regulators—particularly secreted or extracellular molecules—that may serve as both biomarkers and therapeutic vulnerabilities in cancer.

Twenty years ago, the C1q/TNF-related protein family (CTRP1–15) was identified by Wong et al. through a systematic search of human and mouse expressed sequence tag (EST) libraries in the NCBI GenBank database for sequences exhibiting significant homology to the globular C1q domain of adiponectin [7,8]. CTRP family members share a conserved domain architecture comprising an N-terminal signal peptide, a variable region, a collagen-like domain, and a C-terminal globular C1q domain with marked structural similarity to the crystal structure of tumor necrosis factor-α (TNF-α). They can assemble into various multimeric forms, including trimers, hexamers, and higher-order oligomers, and can form either homotrimers or heterooligomers, resulting in distinct biological activities. Similar to the multifunctional adiponectin, CTRPs are involved in diverse processes at the intersection of immunity and metabolism, including the regulation of inflammation, glucose and lipid metabolism, insulin sensitivity, angiogenesis, adipogenesis, and cardiac remodeling [9,10,11,12,13,14]. Since both immune and metabolic pathways contribute to cancer progression, several CTRPs have been implicated in the development and progression of various cancers, including liver, colon, and lung cancers [15]. Among the CTRP family, CTRP6 shows particularly strong associations in a pan-cancer analysis [16]. It is overexpressed in many cancer types, and survival studies indicate that it independently predicts poor prognosis across many tumors, highlighting its promise as both a biomarker and a therapeutic target.

CTRP6 is broadly expressed across both adult and fetal tissues in humans and mice. Its gene expression patterns, protein structure, regulatory mechanisms, and roles in metabolic disorders such as obesity, diabetes, and cardiovascular disease have been summarized in our previous publication [17]. Increasing evidence suggests that CTRP6 also contributes to tumor biology, linking its metabolic and immunomodulatory functions to cancer initiation and progression. Pan-cancer analysis from the TNMplot database [18] reveals that CTRP6 is overexpressed in most cancer types, with particularly high levels in bladder, breast, lung, pancreatic, skin, and uterine cancers (Figure 1A). Similarly, body map data from the GEPIA2 database [19] show that CTRP6 expression is significantly elevated in many tumor tissues compared with normal tissues (Figure 1B). A recent study demonstrated that targeting tumor cell-intrinsic CTRP6 with a biomimetic codelivery system synergistically enhanced ferroptosis and immune activation, thereby improving the efficacy of anti-PD-L1 immunotherapy in lung cancer [20]. These findings underscore the importance of comprehensively examining CTRP6’s roles in cancer. While its relevance to various malignancies, particularly those of the digestive system, has been discussed [21,22], the mechanisms by which CTRP6 contributes to oncogenesis remain poorly defined. In this review, we examine the shared molecular mechanisms and signaling pathways through which CTRP6 regulates tumor biology, emphasizing its diverse roles across different cancer types and its emerging potential as a therapeutic target. We integrate evidence from experimental, computational, and clinical studies to provide mechanistic insights, highlight conflicting findings among cancer types, and discuss new opportunities for therapeutic intervention. A deeper understanding of CTRP6-mediated processes may yield valuable insights into novel therapeutic strategies, particularly in the context of cancer immunotherapy.

## 2. CTRP6 Across Cancer Types

### 2.1. CTRP6 in Hepatocellular Carcinoma

The first cancer type in which CTRP6 was identified to be associated is hepatocellular carcinoma (HCC), the most common primary liver cancer and the second leading cause of cancer-related mortality worldwide. Given the limited treatment options currently available, considerable research in both academic and pharmaceutical settings continues to focus on novel strategies, including immunotherapies, to improve HCC outcomes [23,24]. In 2011, Takeuchi et al. reported that CTRP6 showed focal staining in moderately and well-differentiated HCC nodules, localizing to the cytoplasm of both HCC cells and tumor sinusoidal endothelial cells, but was absent in adjacent non-cancerous liver tissues [25]. These findings suggested that CTRP6 may drive tumor angiogenesis by serving as a paracrine mediator between HCC cells and vascular endothelial cells. This role was further supported by xenograft assays using HepG2 cell clones exogenously expressing CTRP6 [25]. Although overall tumor volumes did not differ significantly between CTRP6-expressing and control xenografts, central hypovascular necrotic regions—commonly present in control tumors—were markedly reduced or absent in CTRP6-expressing tumors, indicating that CTRP6 facilitates neovascularization within HCC.

Beyond its role in angiogenesis, later studies revealed that CTRP6 also contributes to HCC cell survival and invasiveness [26]. Silencing CTRP6 with siRNA not only suppressed Hep3B cell growth but also enhanced caspase-3 activity, thereby promoting apoptosis. In addition, CTRP6 silencing impaired the migratory and invasive capacities of Hep3B cells by reducing AKT phosphorylation. These effects were reversible, as pretreatment with insulin-like growth factor 1 (IGF-1), a known activator of AKT signaling, restored cell migration and invasion while attenuating CTRP6-siRNA–induced apoptosis. Together, these findings indicate that CTRP6 promotes HCC progression through both angiogenesis and AKT-dependent tumor cell survival and invasiveness.

### 2.2. CTRP6 in Lung Cancer

Lung cancer is the leading cause of cancer-related deaths worldwide and represents the cancer type in which CTRP6 has been most extensively investigated. Pathologically, lung cancer is classified into two major subtypes: small cell lung carcinoma (SCLC) and non-small cell lung carcinoma (NSCLC), with lung adenocarcinoma (LUAD) being the most prevalent NSCLC subtype. Since 2019, increasing attention has been given to the role of CTRP6 in lung cancer. Zhang et al. identified CTRP6 as one of the four genes used to construct a prognostic risk model for LUAD [27]. Another study further reported that CTRP6 is upregulated in LUAD tissues and significantly associated with poor prognosis, with computational analyses of TCGA and Oncomine datasets predicting CTRP6 as an independent prognostic marker in lung cancer patients [28].

Mechanistic investigations using A549 cell models revealed that CTRP6 knockdown inhibited proliferation, migration, and invasion while promoting apoptosis [28,29]. Moreover, siRNA-mediated silencing of CTRP6 markedly suppressed xenograft tumor growth compared with controls, whereas CTRP6 overexpression enhanced tumor growth and weight [29]. Together, these findings highlight CTRP6 as a potential therapeutic target in lung cancer.

As CTRP6 has been identified by multiple independent studies as a prognostic marker in patients with LUAD and NSCLC [30,31,32,33], recent research has explored its potential as a therapeutic target. One study employing a cRGD (cyclic arginine-glycine-aspartate)/RBCM (erythrocyte membrane) dual-headed nanocarrier platform co-delivering gemcitabine and silencing CTRP6 expression demonstrated that this strategy amplified ferroptosis in tumor cells via the NRF2/STAT3 signaling pathway [20]. Moreover, it enhanced intratumoral immune activity by promoting M1-like macrophage polarization and CD8+ T-cell recruitment. Importantly, this approach also synergized with PD-L1 monoclonal antibodies to further strengthen anti-lung cancer immunity. These findings highlight the potential of CTRP6 as a promising target in lung cancer immunotherapy.

### 2.3. CTRP6 in Other Types of Cancer

Gastric cancer (GC) ranks as the fourth leading cause of cancer-related deaths worldwide and the fifth most commonly diagnosed cancer [34]. It is commonly classified into two major histological types: intestinal and diffuse. Despite its clinical significance, studies investigating the role of CTRP6 in GC are still limited. In 2019, Qu et al. reported that CTRP6 expression was significantly upregulated (1.59-fold) in GC tissues compared with the peritumoral normal gastric tissues, based on microarray analysis [35]. Functional studies in the human gastric adenocarcinoma cell line AGS demonstrated that CTRP6 knockdown markedly reduced proliferation, colony formation, invasion, and migration, while inducing G2/M cell cycle arrest and apoptosis. These findings suggest that CTRP6 promotes GC cell growth and survival by regulating cell cycle progression and apoptosis. In contrast, the group led by Tamotsu Takeuchi, who first reported CTRP6’s role in HCC, found that treatment with recombinant CTRP6 had no effect on cell growth or Matrigel invasion in three diffuse-type and two intestinal-type GC cell lines, nor on the viability of human gastric epithelial cells [36]. Instead, their study associated CTRP6 with favorable prognosis in diffuse-type GC at distal sites and highlighted its anti-fibrotic properties. They observed focal CTRP6 immunoreactivity in tumor regions with minimal fibrosis, whereas most cancer cells at the invasive front were CTRP6-negative, particularly in areas of extensive fibrosis. These findings suggest that strong fibrotic responses may suppress CTRP6 expression and that restoring or enhancing CTRP6 activity could confer anti-fibrotic effects. These contrasting results from two studies may reflect differences in experimental models and the biological context of CTRP6 activity. While gene knockdown eliminates both intracellular and secreted CTRP6, exogenous recombinant protein may not fully replicate its endogenous functions due to variations in post-translational modification, receptor availability, or microenvironmental cues. Alternatively, CTRP6 may exert distinct intracellular and extracellular roles, such that recombinant protein supplementation cannot compensate for the loss of endogenous expression. Collectively, these findings highlight the complexity of CTRP6 biology and underscore the need to identify its receptor and signaling partners to better elucidate its mechanistic and context-specific functions in gastric cancer.

Tamotsu Takeuchi’s group also demonstrated that recombinant CTRP6 markedly inhibited the proliferation and Matrigel invasion of oral squamous cell carcinoma (OSCC) SAS cells in a dose-dependent manner [37]. Mechanistically, CTRP6 bound to the precursor of the laminin receptor, thereby disrupting laminin-receptor interactions and inhibiting tumor cell invasion. Consistent with these in vitro findings, administration of CTRP6 markedly reduced SAS tumor growth in a xenograft mouse model. In contrast, another study reported that CTRP6 was upregulated in OSCC tissues, and lentiviral shRNA-mediated knockdown of CTRP6 in Cal-27 and SCC-9 cells led to a time-dependent reduction in proliferation, cell-cycle arrest with decreased S-phase and increased G2/M-phase populations, and enhanced apoptosis [38]. Knockdown of CTRP6 also significantly reduced tumor size in nude mice compared with controls. Similarly, elevated CTRP6 expression has been observed in head and neck squamous cell carcinoma (HNSCC), where CTRP6 overexpression markedly promoted proliferation, motility, and angiogenesis of HNSCC cells [39]. These conflicting results suggest that CTRP6 plays dual roles in OSCC/HNSCC, acting as an anti-invasive factor via extracellular receptor interference under certain contexts, while promoting proliferation and survival through intracellular signaling in others. This duality highlights the importance of clarifying CTRP6’s receptor interactions, post-translational modifications, and tumor microenvironmental determinants.

CTRP6 has also been reported to be overexpressed in clear cell renal cell carcinoma (ccRCC), where its expression correlated with key clinicopathological parameters, including TNM classification, cancer stage, metastasis, and tumor grade [40]. In contrast, analysis of two GEO (Gene Expression Omnibus) datasets revealed no significant differences in CTRP6 expression between kidney inflammatory diseases and normal tissues, suggesting that CTRP6 overexpression is more specific to malignant transformation. Collectively, these findings support the potential of CTRP6 as a novel diagnostic and prognostic biomarker for ccRCC [40]. Similarly, high CTRP6 expression has been identified as a predictor of poor prognosis in bladder cancer (BC) patients [41]. Our recent study also found upregulated CTRP6 gene expression in patients with colon adenocarcinoma and uterine corpus endometrial carcinoma [42]. Beyond these cancers, CTRP6 has also been implicated in breast cancer and ovarian cancer [43,44], although current evidence in these tumor types remains limited. A summary of the experimental and clinical evidence regarding CTRP6 expression and its functional roles in diverse cancer types is presented in Table 1, highlighting its reported biological effects and clinical associations.

## 3. Molecular Mechanisms of CTRP6 in Cancer

Understanding the molecular mechanisms by which CTRP6 influences cancer biology is essential for clarifying its role in tumor progression. CTRP6 interacts with diverse signaling networks that regulate fundamental processes, including angiogenesis, ferroptosis, proliferation and apoptosis, migration and invasion, and inflammation (Figure 2).

### 3.1. Angiogenesis

Angiogenesis, the formation of new blood vessels, is critical for cancer growth and progression, providing oxygen and nutrients to support proliferation and enabling invasion and metastasis [45]. It is typically initiated by stimuli such as hypoxia or inflammation, which activate complex signaling cascades regulated by angiogenic activators and inhibitors. Environmental factors like hypoxia, mechanical forces (e.g., shear stress), and various chemical signals further modulate this process, with key mediators including vascular endothelial growth factor (VEGF), fibroblast growth factor (FGF), platelet-derived growth factor (PDGF), angiopoietins (Ang), hepatocyte growth factor (HGF), hypoxia-inducible factor (HIF), insulin-like growth factor (IGF), transforming growth factor-beta (TGF-β), matrix metalloproteinase (MMP), and tumor necrosis factor (TNF) [46]. CTRP6 has emerged as a pro-angiogenic factor in several cancers. In HCC, xenograft assays demonstrated that CTRP6-expressing HepG2 cells exhibited reduced central hypovascular necrosis, indicating enhanced neovascularization. Mechanistically, recombinant CTRP6 increased levels of phosphorylated AKT in cultured vascular endothelial cells, suggesting that CTRP6 promotes angiogenesis, at least in part, through AKT activation [25]. However, the upstream mechanism by which CTRP6 triggers AKT phosphorylation remains unclear. Given its structural similarity to other CTRP family members, CTRP6 may interact with an as-yet-unidentified membrane receptor or cofactor to initiate AKT signaling. Consistently, in HNSCC, CTRP6 overexpression in Detroit562 cells promoted angiogenesis of human umbilical vein endothelial cells (HUVECs), as evidenced by increased tube formation [39]. Activation of AKT stabilizes hypoxia-inducible factor 1α (HIF-1α), which in turn upregulates VEGF, a central mediator of endothelial proliferation and neovascularization [46]. These findings suggest that CTRP6-driven angiogenesis may involve the AKT–HIF-1α–VEGF signaling axis, and the potential involvement of mTOR has not yet been investigated [47]. AKT-dependent mTOR signaling could serve as an intermediary step, amplifying downstream angiogenic responses under hypoxic or metabolically stressed conditions. Future studies identifying the receptor and metabolic context of CTRP6-mediated AKT activation will be critical for defining its precise molecular mechanism in tumor angiogenesis.

### 3.2. Ferroptosis

Ferroptosis is a regulated form of cell death driven by iron-dependent lipid peroxidation. It occurs when reactive oxygen species (ROS) oxidize polyunsaturated fatty acids in membrane phospholipids, leading to the accumulation of phospholipid hydroperoxides, the hallmark of ferroptosis [48]. Unlike apoptosis or necrosis, it is defined by iron overload, unchecked lipid peroxidation, and impaired antioxidant defenses. The system Xc^−^/glutathione (GSH)/glutathione peroxidase 4 (GPX4) axis plays a central role, as reduced cystine uptake or GPX4 inactivation prevents detoxification of lipid peroxides. GPX4-independent mechanisms, such as the FSP1–CoQ10–NAD(P)H pathway, also act as parallel defenses against ferroptosis [49,50]. Many cancers evade ferroptosis by upregulating GPX4, which prevents lethal lipid peroxidation and contributes to therapy resistance. CTRP6 has recently emerged as a suppressor of ferroptosis in cancer. The first evidence came from lung cancer, where CTRP6 knockdown in A549 and H1299 cells reduced GSH, increased iron and malondialdehyde (MDA), and enhanced lipid peroxidation [51]. Mechanistically, CTRP6 inhibits ferroptosis through the xCT/GPX4 axis by promoting SOCS2 ubiquitination and degradation, thereby stabilizing the antioxidant defense system. Therapeutically, a nanoparticle-based system co-delivering siRNA against CTRP6 and gemcitabine significantly enhanced ferroptotic markers and suppressed tumor growth, an effect reversed by ferroptosis inhibition [20]. More recently, in nasopharyngeal carcinoma (NPC), N^6^-methyladenosine (m6A)-hypomethylated CTRP6 was shown to promote radio resistance by stabilizing mitochondria-associated membranes (MAMs) and inhibiting ferroptosis [52]. The m6A RNA modifiers METTL3 (“writer”) and IGF2BP1 (“reader”) recognize a conserved “GGACU” motif within CTRP6 mRNA. In radioresistant NPC cells, downregulation of METTL3 leads to hypomethylation of CTRP6 transcripts, which impairs IGF2BP1 binding and reduces CTRP6 mRNA decay, thereby increasing its expression. Elevated CTRP6 directly interacts with the endoplasmic reticulum (ER) chaperone GRP78/Bip, reinforcing the structural integrity of MAMs and maintaining mitochondrial Fe^2+^ and Ca^2+^ homeostasis. This prevents calcium overload and labile iron accumulation, thereby suppressing ferroptotic cell death and contributing to radioresistance in NPC cells. Together, these studies identify CTRP6 as a ferroptosis suppressor with significant implications for cancer progression and therapy resistance. However, whether CTRP6 functions as a general ferroptosis suppressor—by regulating both the SOCS2–xCT/GPX4 and GRP78–MAMs pathways—or acts in a tumor-type–specific manner remains to be determined. Further investigations across additional cancer types are needed to clarify the cellular contexts in which these mechanisms operate.

### 3.3. Cell Proliferation and Apoptosis

Uncontrolled proliferation and resistance to apoptosis are fundamental hallmarks of cancer [53]. Under normal conditions, cell growth and death are balanced to preserve tissue homeostasis, but tumor cells disrupt this equilibrium by sustaining proliferative signaling and evading apoptosis. This imbalance promotes tumor initiation, progression, and therapy resistance. Thus, uncovering regulators that coordinately modulate proliferation and apoptosis is critical for advancing our understanding of oncogenesis and identifying new therapeutic targets [54]. Multiple studies have implicated CTRP6 as a pro-tumorigenic factor that promotes proliferation and suppresses apoptosis across diverse cancer types. In NSCLC, silencing CTRP6 in A549 cells significantly reduced proliferation, as shown by CCK-8 assay [28]. Similarly, in LUAD, CTRP6 was identified as a direct target of miR-29a-3p, and miR-29a-3p overexpression downregulated CTRP6, markedly inhibiting cell proliferation [55]. Given that CTRP6 has been shown to inhibit ferroptosis, it is plausible that its pro-proliferative effects are partly mediated by maintaining redox homeostasis and preventing ferroptotic cell death. Future studies are needed to determine whether CTRP6-driven suppression of ferroptosis contributes directly to enhanced proliferation and tumor survival. In gastric carcinoma, CTRP6 is frequently overexpressed, and its silencing in AGS cells reduced proliferation while promoting apoptosis [35]. Collectively, these findings highlight CTRP6 as a tumor-promoting factor that drives cancer progression by enhancing proliferative signaling and conferring resistance to apoptosis.

### 3.4. Cell Migration and Invasion

Cell migration and invasion are key processes in cancer progression and metastasis. Migration enables tumor cells to move through the local microenvironment, while invasion allows them to degrade and breach extracellular matrix barriers [56,57]. A central driver of these behaviors is epithelial–mesenchymal transition (EMT), where epithelial tumor cells adopt mesenchymal features that promote motility and invasiveness. These processes are largely mediated by signaling pathways such as PI3K/AKT, MAPK, and TGF-β, and are strongly associated with poor clinical outcomes [58,59]. Identifying regulators of migration and invasion is therefore critical for devising strategies to restrict metastasis. CTRP6 has emerged as a critical regulator of cancer cell migration and invasion across diverse tumor types. In gastric carcinoma, CTRP6 knockdown reduced AGS cell invasion by ~76% in Transwell assays and suppressed migration by >40% in wound-healing assays [35]. Similarly, in lung adenocarcinoma, downregulating CTRP6 in A549 and H1975 cells markedly inhibited both migration and invasion, as demonstrated by wound-healing and Transwell assays [55]. Consistent findings were observed in bladder cancer (5637 and T24 cells), where CTRP6 knockdown suppressed migration and invasion [41]. In HNSCC, CTRP6 promoted migration, invasion, and angiogenesis, effects reversed by miR-29c-3p–mediated suppression of CTRP6 [39]. Collectively, these studies establish CTRP6 as a pro-metastatic factor and highlight its potential as a therapeutic target to restrict tumor dissemination. However, none of these studies have delved into the molecular mechanisms underlying CTRP6-mediated invasion. In particular, it remains unclear whether CTRP6 influences key regulators of EMT, such as the transcription factors *Snail*, *Slug*, and *Twist*, which drive the loss of epithelial characteristics and acquisition of a motile phenotype [60,61]. Similarly, the potential involvement of MMPs, especially MMP-2 and MMP-9—critical enzymes for extracellular matrix degradation—has not been systematically examined in the context of CTRP6 signaling. Moreover, given CTRP6’s nature as a secreted protein, its possible interaction with integrins or laminin receptors to modulate cell–matrix adhesion and cytoskeletal remodeling remains an open question. Elucidating these pathways will be crucial for understanding how CTRP6 orchestrates the metastatic cascade at the molecular level.

### 3.5. Inflammation

Inflammation contributes to cancer initiation and drives its progression through all stages of tumorigenesis [62]. Tumor cells and surrounding stromal and immune cells interact to establish an inflammatory tumor microenvironment (TME), where cellular plasticity enables continuous adaptation in phenotype and function [63]. Within this milieu, inflammatory cells and mediators secrete cytokines, chemokines, and growth factors that enhance proliferation, angiogenesis, and metastasis, while also facilitating immune evasion and supporting tumor cell survival. Recent studies suggest that CTRP6 functions as a positive regulator of inflammatory cascades by targeting both adipocytes and macrophages, thereby promoting obesity-associated inflammation and activating inflammatory gene programs and signaling pathways in mouse bone marrow-derived macrophages (BMDMs) [64,65]. In pan-cancer analyses, CTRP6 expression has been found to correlate positively with monocyte and macrophage infiltration [16]. Furthermore, in uterine carcinosarcoma (UCS), CTRP6 has been implicated in immune regulation, including cytokine signaling pathways, antigen processing and presentation, neutrophil degranulation, and interleukin-mediated immunomodulatory responses [16]. CTRP6 has been shown to induce the expression of proinflammatory cytokines such as IL-6, TNF-α, and IL-1β, which are transcriptionally regulated by NF-κB and STAT3—key drivers of chronic inflammation and immune evasion [66]. Activation of these pathways may enhance cancer cell survival, angiogenesis, and the recruitment of tumor-associated macrophages (TAMs), which, in turn, secrete additional cytokines and growth factors that sustain tumor-promoting inflammation. However, the precise mechanisms through which CTRP6 exerts its effects in cancer remain to be fully elucidated.

## 4. Signaling Mechanisms of CTRP6 in Cancer

Only a few studies have investigated the signaling pathways in the context of CTRP6 and cancer, primarily in hepatocellular carcinoma, lung cancer, and gastric cancer models. Most focus on phenotypic outcomes—such as proliferation, migration, or angiogenesis—without detailed analysis of upstream receptors or downstream effectors. This limits the ability to define causality or generalize findings across cancer types. Nevertheless, available evidence consistently implicates CTRP6 in activating oncogenic signaling. This review integrates current data to highlight CTRP6 as a potential pathway regulator and identify key gaps for future mechanistic and translational studies.

### 4.1. PI3K/AKT Pathway

The PI3K/AKT signaling pathway is one of the most commonly dysregulated intracellular cascades in cancer, with critical roles in promoting tumor cell proliferation, survival, invasion, and metastasis [67,68]. Dysregulation occurs through multiple mechanisms, including genetic mutations, epigenetic changes, altered microRNA regulation, and abnormal phosphorylation events [69,70]. These disruptions not only drive tumorigenesis but also contribute to therapeutic resistance, underscoring the pathway’s importance as a central hub in cancer biology. Emerging evidence indicates that CTRP6 is closely linked to PI3K/AKT pathway activity. As a secreted protein, CTRP6 is thought to interact with cell surface receptors to initiate downstream intracellular signaling; however, these receptors have yet to be definitively identified. Although interactions with potential binding partners such as AdipoR1, laminin, and RXFP1 have been reported [37,71,72], the existence of a specific CTRP6 receptor remains unconfirmed and requires validation using receptor knockout models under physiological conditions. Multiple studies demonstrate that CTRP6 activates AKT, thereby influencing a wide range of oncogenic processes (Figure 3). In HCC, recombinant CTRP6 significantly increased AKT phosphorylation in human liver sinusoidal microvascular endothelial cells, promoting tumor angiogenesis [25]. The C-terminal C1q domain of CTRP6 appears to be essential for this activation. The AKT–mTOR–p70S6K pathway is well established in embryonic vascular development and pathological angiogenesis [47]. In tumor cells, activation of PI3K/AKT/mTOR signaling stimulates VEGF secretion through both HIF-1–dependent and –independent mechanisms, while also regulating other angiogenic mediators such as nitric oxide and angiopoietins. CTRP6 may exert its pro-angiogenic effects through this signaling axis, although direct evidence is still limited. Consistent with this, silencing CTRP6 in Hep3B cells suppressed AKT activation and reduced proliferation, migration, invasion, and survival, effects that were restored by the AKT activator IGF-1 [26]. Together, these findings strongly suggest that CTRP6 promotes tumor progression, at least in part, through activation of PI3K/AKT signaling. However, outside of HCC, evidence for CTRP6-driven PI3K/AKT activation in digestive system tumors and other cancer types remains scarce, highlighting the need for further investigation.

### 4.2. MAPK (MEK/ERK) Pathway

The mitogen-activated protein kinase (MAPK) pathway is a highly interconnected signaling network frequently implicated in oncogenesis, tumor progression, and therapy resistance. The cascade proceeds from MAPKKKs (e.g., RAF) to MAPKKs (MEK1–7), and ultimately to MAPKs. The three major MAPK families include ERKs (ERK1/2), JNKs (JNK1–3), and p38 MAPKs (p38α, β, γ, δ) [73]. Among them, MEK and ERK1/2 regulate key processes such as cell survival, proliferation, and differentiation through phosphorylation of diverse targets. ERK1/2, in particular, acts on substrates in multiple cellular compartments; in the nucleus, it activates transcription factors including CREB, c-Myc, and NF-κB, making it an important therapeutic target. Dysregulation of MAPK signaling is common in cancer, and numerous kinases within this pathway have been pursued for therapeutic intervention. CTRP6 has been reported to regulate the MAPK pathway in several cancer types (Figure 4). In lung adenocarcinoma A549 cells, silencing CTRP6 significantly reduced MEK and ERK phosphorylation without altering their protein expression [28]. In gastric and other digestive tract cancers, CTRP6 may promote tumor progression by activating the MAPK/ERK1/2 pathway, enhancing inflammatory cytokine production such as IL-1β, IL-6, and TNF-α [21]. Since these cytokines are transcriptionally regulated by NF-κB and STAT3, it is likely that CTRP6-driven ERK activation intersects with these pathways to sustain tumor-associated inflammation.

As a secreted ligand, CTRP6 may act as a central hub integrating both the PI3K/AKT and MEK/ERK signaling pathways, thereby promoting angiogenesis, proliferation, migration, invasion, and inflammatory reprogramming within the tumor microenvironment. This coordinated activation positions CTRP6 as a key modulator linking oncogenic and inflammatory signaling networks.

## 5. Therapeutic Potential of CTRP6 in Cancer

CTRP6 has emerged as a promising therapeutic target in multiple cancer types due to its multifaceted roles in tumor progression, angiogenesis, ferroptosis, and therapy resistance. In lung cancer, CTRP6 promotes tumor proliferation and metastasis, while its knockdown induces ferroptosis via the xCT/GPX4 axis through SOCS2 ubiquitination and degradation [51], suggesting that targeting CTRP6 could sensitize tumors to ferroptosis-based therapies. In nasopharyngeal carcinoma, m6A-hypomethylated CTRP6 drives radio resistance by stabilizing MAMs and inhibiting ferroptosis through the CTRP6–GRP78 axis [52], highlighting its role in therapy evasion. Translationally, innovative strategies such as nanoparticle-mediated co-delivery of siRNA against CTRP6 and chemotherapeutic agents like gemcitabine have demonstrated significant tumor suppression in preclinical models by amplifying ferroptotic cell death [20].

Despite these promising findings, several challenges remain before CTRP6-based therapies can be translated into clinical applications. CTRP6 functions appear to be context-dependent, exhibiting both pro- and anti-tumor effects in different cancer types, which underscores the need for careful patient stratification and molecular characterization. Its receptor and downstream mediators also remain poorly defined, limiting the rational design of selective inhibitors. From a pharmacological standpoint, CTRP6 is a secreted circulating protein, making it theoretically druggable through antibody-based neutralization or ligand–receptor blockade; however, such approaches would require high target specificity to minimize off-target effects. Systemic inhibition may disrupt normal physiological functions, as CTRP6 is involved in metabolism, vascular remodeling, and tissue homeostasis.

Delivery also represents a major obstacle—achieving tumor-specific targeting of macromolecules remains challenging due to rapid clearance, limited tissue penetration, and immune recognition. Future therapeutic efforts should therefore prioritize the development of targeted delivery systems, such as tumor-directed nanoparticles, antibody–drug conjugates, or engineered extracellular vesicles, to enhance efficacy while reducing systemic toxicity. Integrating in vivo modeling, multi-omics profiling, and biomarker-guided strategies will be essential to determine the feasibility and safety of CTRP6 inhibition. Ultimately, elucidating how CTRP6 interfaces with oncogenic and immune pathways will guide the design of precision therapies and improve cancer treatment outcomes.

## 6. Conclusions 

CTRP6 is a multifunctional regulator of cancer biology, driving angiogenesis, proliferation, migration, inflammation, therapy resistance, and ferroptosis suppression through pathways such as PI3K/AKT and MEK/ERK. Current evidence supports its largely pro-tumorigenic role in cancers such as hepatocellular carcinoma, lung cancer, and clear cell renal cell carcinoma; however, conflicting results in gastric and head and neck cancers suggest context-dependent functions that require further clarification.

Despite growing evidence linking CTRP6 to tumor progression, major knowledge gaps remain. The specific receptor or binding partners mediating its signaling have yet to be identified, and the phosphorylation sites and post-translational modifications governing its downstream activation remain uncharacterized. Furthermore, how CTRP6 interfaces with immune signaling—particularly NF-κB, STAT3, and tumor-associated macrophage polarization—remains poorly understood. Most studies are still confined to in vitro systems, with limited validation in in vivo or clinical settings.

Future research should focus on identifying CTRP6 receptors, mapping phosphorylation-dependent signaling events, and investigating its role in immune evasion and tumor–microenvironment crosstalk. It will also be important to test CTRP6 inhibition using siRNA, neutralizing antibodies, or small molecules in animal models. Integrating multi-omics profiling with functional and translational studies will be crucial to determine CTRP6’s clinical potential. A deeper understanding of these mechanisms will not only resolve existing inconsistencies but also accelerate the development of CTRP6-based diagnostic and therapeutic strategies in precision oncology.

## Figures and Tables

**Figure 1 cancers-17-03409-f001:**
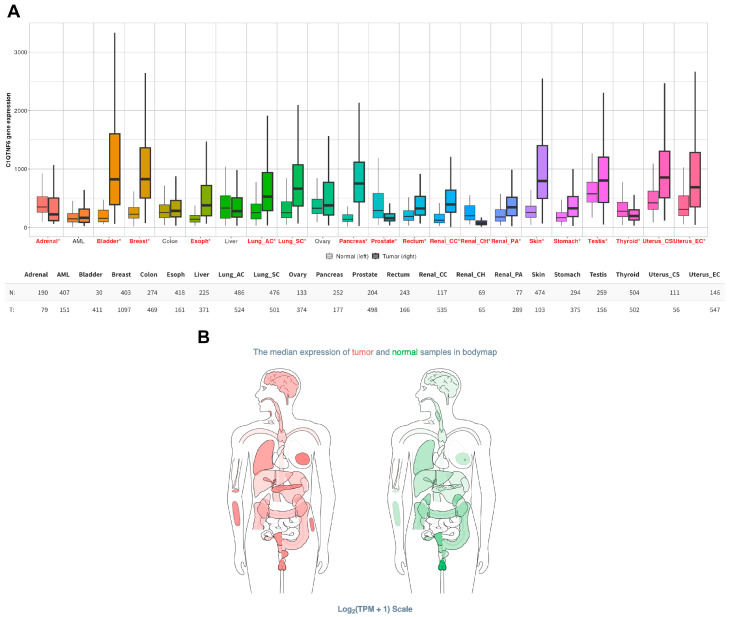
CTRP6 expression across human cancers. (**A**) Pan-cancer box plot illustrating CTRP6 expression in tumor and matched normal tissues generated from RNA-Seq data in the TNMplot database (https://tnmplot.com, accessed on 15 August 2025). Red asterisks (*) indicate significant differences (Mann–Whitney *p* < 0.05) with expression > 10 in either tumor or normal samples. CTRP6 shows marked overexpression in bladder, breast, lung, pancreatic, skin, and uterine cancers, indicating a potential oncogenic role in these malignancies. (**B**) Median CTRP6 expression in tumor and normal tissues across different organs, as illustrated in the bodymap from the GEPIA2 database (http://gepia2.cancer-pku.cn/, accessed on 15 August 2025).

**Figure 2 cancers-17-03409-f002:**
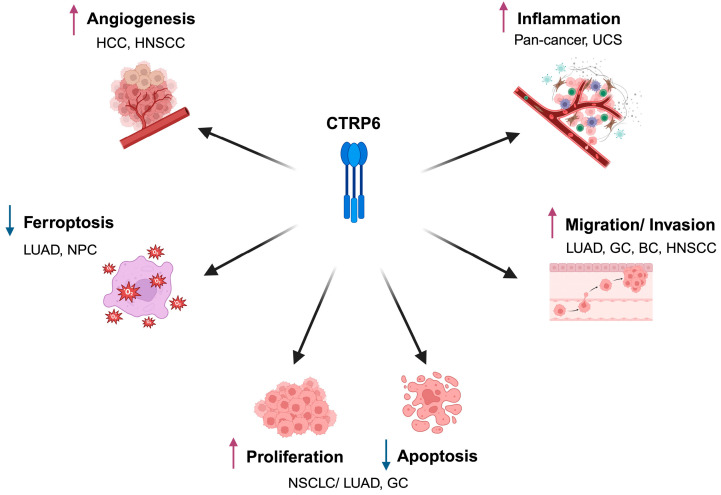
CTRP6 is involved in diverse processes across different cancer types, including angiogenesis, ferroptosis, proliferation and apoptosis, migration and invasion, and inflammation. The figure integrates experimentally validated molecular mechanisms through which CTRP6 regulates tumor biology across the different cancer types discussed in this review. Cancer types: HCC, hepatocellular carcinoma; HNSCC, head and neck squamous cell carcinoma; LUAD, lung adenocarcinoma; NPC, nasopharyngeal carcinoma; NSCLC, non-small cell lung cancer; GC, gastric cancer; BC, bladder cancer; UCS, uterine carcinosarcoma. ↑ indicates promotion; ↓ indicates inhibition. Created in BioRender. Lei, X. (2025) https://BioRender.com/mrf7urg (accessed on 25 August 2025).

**Figure 3 cancers-17-03409-f003:**
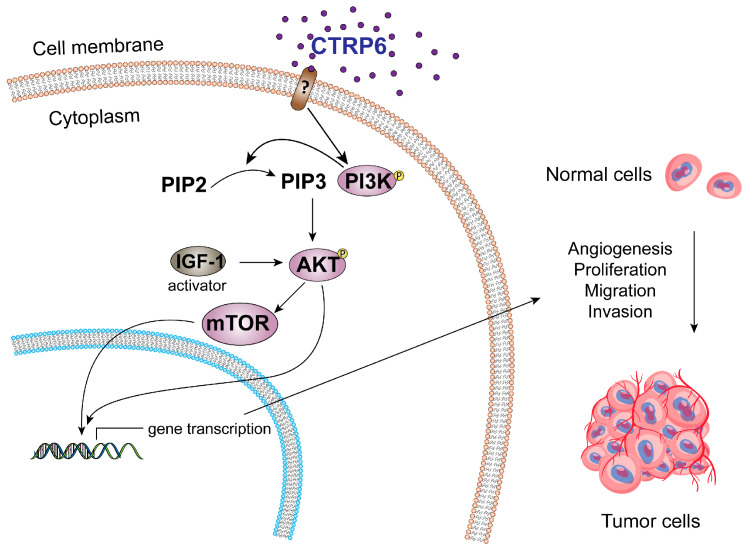
CTRP6 affects angiogenesis, proliferation, migration, and invasion through activation of the PI3K/AKT signaling pathway during cancer progression.

**Figure 4 cancers-17-03409-f004:**
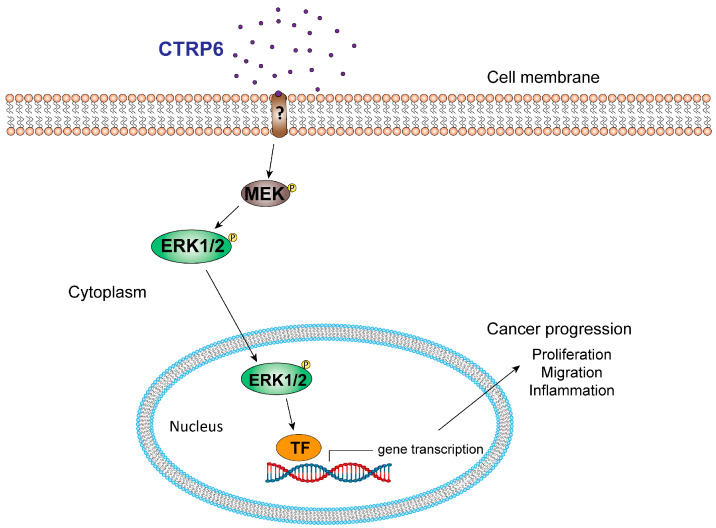
CTRP6 regulates cell proliferation, migration, and inflammation during cancer progression via the MEK/ERK signaling pathway. TF: transcription factors.

**Table 1 cancers-17-03409-t001:** Summary of CTRP6 expression and functional roles across various cancer types. CTRP6 expression patterns, biological effects, and clinical implications are summarized based on available experimental and clinical studies.

Cancer Type	CTRP6 Expression	Role in Tumor Biology	Clinical Relevance
Hepatocellular carcinoma (HCC)	Upregulated	Promotes angiogenesis and survival via AKT signaling [25,26]	Associated with tumor vascularization
Lung cancer (LUAD/NSCLC)	Upregulated	Enhances proliferation, migration, invasion; inhibits ferroptosis [20,28,29]	Poor prognosis, therapy resistance
Gastric cancer (GC)	Upregulated/context-dependent	Promotes or suppresses proliferation and fibrosis depending on subtype [35,36]	Prognostic marker; conflicting results
Oral/Head and neck squamous cell carcinoma (OSCC/HNSCC)	Variable	Dual role: suppresses invasion [37] or promotes proliferation [38,39]	Context-dependent behavior
Clear cell renal cell carcinoma (ccRCC)	Upregulated	Promotes metastasis and progression [40]	Correlates with TNM stage and tumor grade
Bladder cancer (BC)	Upregulated	Enhances migration and invasion [41]	Predicts poor prognosis

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
