# Peer review of "CTRP6 in Cancer: Mechanistic Insights and Therapeutic Potential"

_cancers, 2025, doi:10.3390/cancers17213409_

Round 1

Reviewer 1 Report

Comments and Suggestions for Authors

the research topic is very interesting and could bring to medicinal and pharmaceutical society especially regarding cancers treatment. the authors focus their research on specific chemical compound CTRP6 which have specific role  in various cancer types. the selected aim of the study was fulfilled in great extent.

the article is easy to read and its keep focus on the aim of the study through whole article. the introduction explains importance of the compound and its predicted role in cancers development. 

the authors analyze specific cancers type with its ethology and morbidity. 

following paragraph analyses cell processes such as ferroptosis, proliferation and apoptosis. 

the therapeutic effect is also well described. 

the number of figures and tables are excellent and give opportunity to the reader to make conclusion by itself.

the references are adequate for the article and support the research framework.

the conclusion summarizes findings and underlines the most important one.

 but before fulfilling the minimum requirements for publication following concerns must be improved.

the authors should add what was the research topic, which scientific database was analyzed how many articles were collected and what was the key topic for including in review?

In my opinion the authors should give structure of CTRP6, its corresponding chemical name and potential active site.

 the paragraph 5 is shallow, and the authors should extend with key responses regrading therapeutic potential of CTRP6 and how its future prognosing.

Reviewer 2 Report

Comments and Suggestions for Authors
  1. The title is clear and concise.
  2. In the abstract, the authors should highlight the pivotal role of CTRP6 in a range of cancers, emphasizing its promising potential as both a biomarker and a therapeutic target.
  3. The introduction is brief, and the flow could be improved by first discussing the limitations of current cancer therapies and the need for novel targets, followed by the introduction of CTRP6 as a promising candidate. While discussing limitations of conventional therapies, cite the following relevant article: 10.1080/09205063.2023.2294541
  4. The labels and colors in Figure 1 should be made clearer, and the legend should briefly explain the most important differences, including which cancers show the strongest changes and why that matters.
  5. In subsection 2, incorporating a comparative table could greatly enhance clarity. This table could effectively illustrate the various cancer types, the expression levels of CTRP6, whether its role is to promote or suppress, and its clinical relevance. This approach would provide a clear and organized overview for readers.
  6. Under subheading 3, highlight how these mechanisms connect to each other, instead of treating them separately. A figure for illustrating the molecular mechanism will be helpful.
  7. The signalling section explains the PI3K/AKT and MAPK/ERK pathways well, but it gives too much general background and not enough focus on CTRP6 itself.
  8. The therapeutic potential section should focus more on the most promising strategies, the main challenges, and how the laboratory findings could be translated into patient care.
  9. The conclusion should address the research gap and future direction more clearly.
  10. Incorporating detailed conflicting findings across cancer types, exploring unknown mechanisms such as the receptor for CTRP6, and tackling the challenges of moving from preclinical studies to clinical applications will not only enrich the review but also inspire greater strides in our fight against cancer.

Reviewer 3 Report

Comments and Suggestions for Authors

Major Concerns: 
1. Limited translational insight
The review focuses heavily on intracellular signaling without addressing the broader therapeutic implications. There is little discussion of whether CTRP6 is druggable, how its inhibition could be achieved, or what off-target or systemic effects might arise. The therapeutic section would gain value from a more realistic assessment of pharmacological feasibility, delivery challenges, and potential side effects.

2. Fragmented treatment of ferroptosis
Although ferroptosis is emphasized as a major mechanism, the discussion feels fragmented. The text describes two distinct regulatory routes—SOCS2–xCT/GPX4 and METTL3–IGF2BP1–GRP78–MAM—but stops short of explaining how they relate to one another or under what cellular contexts they occur.
The section could be strengthened by integrating these pathways conceptually and clarifying whether CTRP6 acts as a general ferroptosis suppressor or in a tumor-type–specific manner.

Section-by-Section Evaluation:
3. Figure 1 – CTRP6 Expression Across Human Cancers
The statistical presentation is minimal, relying only on a Mann–Whitney test without providing information on sample size, dataset source (e.g., TCGA, GTEx), or correction for multiple testing. The biological meaning of the expression differences is not explored, and the figure legend does not explain what the expression threshold (“>10”) represents. Overall, the figure lacks interpretive depth and does not link CTRP6 expression to any clinical or immunological parameters.

4. Section 2 – CTRP6 Across Cancer Types
This section mainly restates published data without building a unifying model. The contrasting roles of CTRP6 across cancers are mentioned but not explained. The review would benefit from a synthesis that considers factors such as tumor metabolism, microenvironmental context, or cytokine signaling, which might shape CTRP6’s behavior.
For HCC and lung cancer, the discussion relies heavily on earlier studies and overlooks recent insights from high-throughput or immunotherapy-related analyses.
The inconsistencies in gastric and oral cancers are noted but not analyzed—possible methodological differences (cell lines, protein dosage, or subtype variations) should be acknowledged to clarify why results diverge.
5.
Figure 2
The figure appears more schematic than mechanistic.
It lists biological processes rather than illustrating how CTRP6 coordinates or links them. Cross-talk among angiogenesis, inflammation, and ferroptosis is not visualized. The legend lacks details about how the figure was constructed or which data it represents.
6.
Section 3.1 – Angiogenesis
This part reiterates earlier findings on HCC and HNSCC with little new interpretation. The AKT–HIF-1α–VEGF axis is presented as a central mechanism, but the upstream trigger of AKT activation remains unspecified. The possible involvement of mTOR or metabolic signaling is mentioned but not explored. The section would be more compelling if it clarified whether CTRP6 interacts directly with a receptor or cofactor to initiate AKT phosphorylation.
7.
Section 3.2 – Ferroptosis
Conceptually strong, but the narrative feels disconnected. Two different regulatory pathways are discussed without integration, and the context-dependence of each mechanism is unclear. The terminology (MAMs, m6A, GRP78) is used without sufficient explanation, making it difficult for non-specialists to follow. A brief, cohesive model summarizing how CTRP6 prevents ferroptosis would significantly improve readability.
8.
Section 3.3 – Proliferation and Apoptosis
This subsection mostly repeats experimental findings without deeper interpretation. It would benefit from showing how CTRP6-driven proliferation is mechanistically linked to its effects on ferroptosis or apoptosis resistance, possibly through common pathways like PI3K/AKT or NF-κB.
9. Section 3.4 – Migration and Invasion
The evidence presented supports CTRP6’s role in promoting metastasis, yet the mechanistic basis remains superficial. There is no mention of whether CTRP6 influences EMT-related transcription factors, matrix metalloproteinases, or integrin signaling—all critical mediators of invasion. This limits the explanatory power of the section.

10. Section 3.5 – Inflammation
This is the weakest part of the manuscript. Much of the evidence comes from non-cancer models (e.g., adipocytes, macrophages), and the link to tumor-associated inflammation is not clearly drawn.
The discussion would be more meaningful if it connected CTRP6 activity to known inflammatory signaling pathways within tumors, such as NF-κB or STAT3, and to immune evasion mechanisms in the tumor microenvironment.
11. Section 4.1 – PI3K/AKT Pathway
The introduction to this pathway is solid, but the role of CTRP6 within it remains vague. The review suggests CTRP6 “may interact with cell surface receptors” without proposing candidates or supporting evidence. Because this interaction could be central to CTRP6’s function, identifying or even discussing potential receptors (such as AdipoR1/R2, CD36, TLR4, or integrins) would strengthen the section.
There is also an opportunity to connect this pathway to ferroptosis resistance, autophagy, or metabolic control, which is not currently addressed.
12. Section 4.2 – MAPK (MEK/ERK) Pathway
The description of MAPK signaling is accurate but generic. Only lung adenocarcinoma and gastric cancer are mentioned, which limits the scope. The statement that CTRP6 activates inflammatory cytokines via ERK is interesting but lacks mechanistic depth.
The potential interaction with NF-κB or STAT3 pathways should be considered, as it could explain CTRP6’s influence on tumor-associated inflammation. A brief note on possible concurrent activation of AKT and ERK pathways would also help unify the signaling perspective.
13. Section 5 – Therapeutic Potential of CTRP6
The therapeutic discussion is underdeveloped. It refers to preclinical work but does not evaluate feasibility or safety in practical terms.
No information is provided about potential therapeutic modalities or delivery strategies. The tone also assumes CTRP6 is strictly oncogenic, overlooking evidence that CTRP6 may have context-dependent protective roles in certain tumors. A balanced consideration of these dual functions would enhance the credibility of the review.
14. Section 6 – Conclusion and Future Directions
The conclusion is concise but lacks specificity. It would be helpful to highlight clear priorities for future research, such as identifying CTRP6 receptors, mapping phosphorylation sites in downstream signaling, exploring its role in immune evasion, and testing CTRP6 inhibition in vivo. The current closing paragraph feels generic and could be more directive in outlining the next steps for the field.

Reviewer 4 Report

Comments and Suggestions for Authors

While CTRP6 has emerged as an interesting candidate in cancer biology, multiple recent reviews have already addressed this topic, including its roles in digestive cancers, signaling pathways (PI3K/AKT, MAPK), ferroptosis, and tumor immunity. The manuscript largely consolidates these published findings without providing significant new perspectives, conceptual frameworks, or forward-looking insights.

The paper reads primarily as a descriptive compilation of existing studies rather than a critical synthesis. For a high-impact journal, the review needs to move beyond summarization and instead identify gaps, controversies, and directions for future research.

The review reiterates findings from single studies (e.g., gastric cancer, OSCC/HNSCC) without sufficiently discussing the contradictory nature of results or proposing reasons for discrepancies (e.g., cell-type specificity, tumor microenvironment, isoform/variant expression).

Mechanistic sections (angiogenesis, ferroptosis, inflammation, etc.) are largely descriptive and lack integration into a unifying model of CTRP6 biology in cancer.

The “Future Directions” section is short and does not provide a strong forward-looking vision. Suggestions such as exploring CTRP6 as a therapeutic target, or investigating its dual tumor-promoting/tumor-suppressive roles, remain generic.

The review would benefit from a perspective-based analysis, for example:

How CTRP6 could integrate with metabolic and immune checkpoint pathways in specific tumor contexts.

Whether CTRP6 might serve as a context-dependent companion biomarker for immunotherapy response.

Gaps in preclinical models (knock-in/out mice, organoids) and translational limitations.

The manuscript includes only two schematic figures (CTRP6 expression and signaling). However:

Figures are basic and do not provide high-level conceptual clarity (e.g., interaction maps, therapeutic opportunities, context-dependent models).

For a complex, multifunctional protein like CTRP6, additional integrative figures (e.g., pan-cancer overview, contradictory roles in different tumor types, network diagrams) would greatly improve readability and impact.

Several sections reiterate general cancer biology (angiogenesis mediators, ferroptosis pathways, MAPK/PI3K/AKT signaling) without clearly linking back to CTRP6-specific insights. This dilutes the originality of the review.

Some studies are described in detail (cell lines, assays) but without critical interpretation of their broader implications or limitations.

Comments on the Quality of English Language

Overall writing is clear and well-organized, but it often follows a study-by-study reporting style. The paper would benefit from more comparative synthesis tables (e.g., “CTRP6 roles across cancers,” “Effects on signaling pathways in different contexts”) to highlight consistencies and contradictions.

An acronym list is included, which is helpful, but clinical aspects (diagnostic/prognostic use, therapeutic targeting) could be better highlighted in a dedicated translational medicine section.

Round 2

Reviewer 1 Report

Comments and Suggestions for Authors

the authors made an effort to clarify concerns and respond to them. the authors responses improve article quality and now it fulfills the minimum requirements for publication and could bring benefit to the scientific community.

Reviewer 2 Report

Comments and Suggestions for Authors

May be accepted for publication.

Reviewer 3 Report

Comments and Suggestions for Authors

I greatly enjoyed reading the revised version of the manuscript entitled “CTRP6 in Cancer: Mechanistic Insights and Therapeutic Potential.”
The authors have carefully addressed the previous comments and substantially improved the scientific quality and clarity of the manuscript.

I would like to congratulate the authors for their efforts in producing a coherent, up-to-date, and well-written paper.

In my opinion, the current version of the manuscript is scientifically sound and well-prepared, and therefore it can be accepted in its present form.

Best regards,

Reviewer 4 Report

Comments and Suggestions for Authors

The Authors have answered to the majority of my previous comments albeit their answers in most cases are rather generic and the effective alterations in the manuscript are barely sufficient

Comments on the Quality of English Language

OK